# Proposed Long Short-Term Memory Model Utilizing Multiple Strands for Enhanced Forecasting and Classification of Sensory Measurements

**Sotirios Kontogiannis** [1,*] **, George Kokkonis** [2] **and Christos Pikridas** [3]

1. Laboratory Team of Distributed Microcomputer Systems, Department of Mathematics, University of Ioannina, 45110 Ioannina, Greece
2. Department of Information and Electronic Engineering, International Hellenic University, 57001 Thessaloniki, Greece; gkokkonis@ihu.gr
3. School of Rural and Surveying Engineering, Aristotle University of Thessaloniki, 54124 Thessaloniki, Greece; cpik@topo.auth.gr
* Correspondence: skontog@uoi.gr

**Abstract:** This paper presents a new deep learning model called the stranded Long Short-Term Memory. The model utilizes arbitrary LSTM recurrent neural networks of variable cell depths organized in classes. The proposed model can adapt to classifying emergencies at different intervals or provide measurement predictions using class-annotated or time-shifted series of sensory data inputs. In order to outperform the ordinary LSTM model's classifications or forecasts by minimizing losses, stranded LSTM maintains three different weight-based strategies that can be arbitrarily selected prior to model training, as follows: least loss, weighted least loss, and fuzzy least loss in the LSTM model selection and inference process. The model has been tested against LSTM models for forecasting and classification, using a time series of temperature and humidity measurements taken from meteorological stations and class-annotated temperature measurements from Industrial compressors accordingly. From the experimental classification results, the stranded LSTM model outperformed 0.9–2.3% of the LSTM models carrying dual-stacked LSTM cells in terms of accuracy. Regarding the forecasting experimental results, the forecast aggregation weighted and fuzzy least loss strategies performed 5–7% better, with less loss, using the selected LSTM model strands supported by the model's least loss strategy.

**Keywords:** deep learning; forecasting algorithms; classification algorithms; neural networks; recurrent neural networks

**MSC:** 68T05

## 1. Introduction

Deep learning has revolutionized data analysis by automating feature extraction and enabling processing time series high-dimensional or even unstructured sensory data, offering classifications and forecasts required for decision making and planning [1]. Unlike traditional classification methods that rely on predefined threshold values or weak AI, task-specific Machine Learning (ML) models for classification and prediction are supported by regression, Support Vector Machines, and Decision Trees. On the other hand, Deep Learning (DL) models excel in supervised, unsupervised, and reinforcement learning utilizing big data, supporting automated decision processes and transfer learning via suggestions and indications [2–4].

DL models like Artificial Neural Networks (ANNs-NNs), Convolutional Networks (CNNs) combined with ANNs, and Recurrent Neural Networks (RNNs) disseminate data to patterns and learn from patterns, making them particularly effective for complex sensory environments. This adaptability has been pivotal in industries where artificial neural networks (ANNs) have improved forecasting and classification accuracy and minimum loss over ARIMA and ARMA models in periodic and real-time data environments [5,6]. That leaves space for more elegant ARMA-NN models to perform better than NN models [7].

Recent advancements in DL models significantly impacted incident predictions via the classification and forecasting of sensory measurements. In particular, these leverage either Recurrent Neural Network (RNN) [8] models, Long Short-Term Memory (LSTMs), Gate Recurrent Units (GRUs) [9], classification fuzzy set transformations [10,11], or convolution filtering for feature extraction combined with neural networks [8,12]. Artificial neural networks treat inputs independently. Data inputs are processed as a batch of data where batch attributes can be inferred, and inter-entity dependencies and correlations are not considered. Consequently, neural networks can classify sensory batches and provide batch forecasts.

Recent studies of stranded-NN [13] and fuzzy stranded-NN [11] models that use sets of multiple neural networks managed in some cases to achieve 1-5% better classification accuracy rover dual layer-stacked LSTM models. Nevertheless, fully connected NN's alone cannot outperform the LSTM models' prediction and classification capabilities and generally perform less well in terms of precision and accuracy [14]. Furthermore, generative adversarial networks (GANs), consisting of a generator network responsible for sensory data generation and a discriminator network designed to classify such generated data, are a type of neural network used for unsupervised machine learning such generatepurposes [15]. Additionally, GANs utilize parallel neural networks by leveraging parallelization techniques, thus enabling augmented classification processes.

Recurrent Neural Networks (RNNs) are deep learning models that include internal memory, enabling them to process sequential data and identify dependencies or patterns. RNNs consider the temporal order of inputs, thus making them suitable for tasks involving temporal predictions and forecasting. However, deep RNN problems in the differentiation of short-term and mainly long-term patterns due to vanishing gradient limitations restrict their ability to perform adequately, showing similar loss characteristics with NN networks over training epochs [16,17].

Several improved variants have been developed to deal with traditional Recurrent Neural Networks (RNNs) issues. These include the Long Short-Term Memory (LSTM) model, which was designed to tackle the vanishing gradient problem [18], as well as bidirectional LSTM (BiLSTM) or GRU (BiGRU) models [19]. Additionally, Gated Recurrent Unit (GRU) models offer a more efficient training and inference alternative than LSTM networks [20]. Other variations include bidirectional GRUs [21] and Bayesian RNNs, which are probabilistic models that incorporate Bayesian inferences to account for uncertainty in the network's predictions [22,23], along with several others.

LSTM and GRU models pertain to significant performance capabilities in terms of accuracy, providing minimum loss compared to RNN and NN models [17]. Comparing GRU and LSTM models in terms of accuracy and loss, there are several studies in the literature showing that both models perform similarly [24,25], with the LSTM models slightly outperforming GRUs for long-range dependencies [25] or complex contexts [26]. That is why the authors primarily used LSTM models for the cross-comparison tasks of their following proposition of the stranded LSTM model.

Additionally, bidirectional LSTM and GRU models can be used in some cases of sensory predictions [27,28] and provide minimal losses for forecasting problems due to

their ability to capture bidirectional features of time series data [29]. They are particularly beneficial in natural language processing (NLP) classification and data generation tasks [30,31].

Bidirectional RNNs such as LSTM (BiLSTM) or GRU (BiGRU) models can offer independent forward and backward (LSTM or GRU) processing time domain information as a window time frame, offering the aggregations of sequences from start to finish and finish to start [19]. This way, both past-to-future and future-to-past context is acquired. Such models yield worse results, compared to LSTM models, for forecasting tasks [15] and can operate successfully on classification tasks. Such models or combinations of such models are capable of temporal feature extraction or denoising/processing entire windows of time series rather than real-time forecasting [32]. The authors argue that while bidirectional models such as BiLSTM are fair classifiers, utilizing temporal frames of both past and future context, their application in forecasting tasks can be misleading. These models use a reverse chronological order of events during training, effectively aggregating information from future time steps that would not be available in real-time predictions. Consequently, this may result in an overestimation of the model's predictive performance and reduce the applicability that such models can offer to actual forecasting or classification problems since they confuse their temporal data events, utilizing reverse chronological order sequences and aggregating them with the actual temporal flow, thus leading to a less accurate results [8,33].

Convolutional Neural Networks (CNNs) process sensory data, generating patterns and extracting locality features (spatial, sensing, and hybrid) [34]. Two-dimensional CNNs are commonly used for image or image stream classification and object detection or to classify location-aware sensory-grid values. Three-dimensional CNNs are mainly used for 3D image and volumetric sensory classification. In contrast, one-dimensional CNNs are commonly used for forecasting time series of sensory vector values for per asset (single point) fault detection and predictive maintenance [35]. Two-dimensional CNN models combined with NN networks may sometimes outperform existing RNN models [36,37], especially when sensory and sound spectrograms are involved. Some studies show accuracy improvements of 1–2% for 1D-CNN LSTM models in data series classification tasks compared to LSTM models in terms of accuracy and F1-score [38].

Hybrid CNN LSTM models seem to perform better [39] or are mentioned as a promising approach for improving time series data prediction [40], concerning their LSTM model counterparts, depending on the use case. As also mentioned in [41], CNN-LSTM and BO-CNN-LSTM models can outperform LSTM models by more than 7–10% in some cases. LSTM and GRU models present almost similar accuracy results in terms of forecasting, with cases of 3–4% improvement in accuracy predictions, as enforced by GRU models [17,42]. As mentioned by [43], there is at least 1% *MAPE* improvement when using GRU models compared to LSTM and 33% faster training. On the other hand, neural networks using wavelets decompose mainly close-to-real-time or real-time signals into time-frequency components, capturing both transient and steady-state features instead of CNNs for feature decomposition and extraction, mainly used for classification tasks and offering significant accuracy results [44,45].

This paper introduces a novel LSTM model that integrates multiple LSTM models, referred to as strands, into a single entity. By employing ensemble methods, the model combines predictions from these various LSTM models, which serve as weak learners, focusing on automated, distributed, and multi-modal deep learning processes [46]. The proposed model is called stranded LSTM and utilizes automated selection processes called strategies for providing a distributed prediction or classification result based on multiple

class differentiated inferences, provided as an aggregation output by the selected strategy. The final prediction or classification is generated by weight aggregating the inference outputs from the selected class LSTM model's inferences of different cell depths. This approach mitigates overfitting and provides automated predictions of variable granularity for high dynamic range tasks and tasks that include longer and smoother patterns, thus enhancing performance compared to any individual LSTM model.

The paper is organized as follows: Section 2 presents the proposed stranded LSTM model and its implemented inference selection strategies. Section 3 presents the authors' model experimentation in (a) forecasting case scenario and (b) classification case scenario. Section 4 discusses the results of the different model selection strategies and model limitations, and Section 5 concludes the paper.

## 2. Materials and Methods

### 2.1. Proposed Stranded LSTM Model

The authors propose a new deep learning model called stranded LSTM. This model utilizes multiple strands of LSTM models as a single entity. The coexisting LSTM models (strands) in the stranded LSTM model are further subdivided into four classes that differentiate on the LSTM model's memory depth. This way, a multi-granularity hierarchical network is provided for long- and short-term forecasting [47]. The term granularity indicates the ability of each network to adapt to time series changes. Therefore, high granularity classes focus on short time interval changes (audio signals, IoT sensors), while low granularity classes prioritize capturing broader temporal patterns (ECG, climate data). Depending on the LSTM cell depth, Class 1 includes LSTM strands with LSTM models consisting of a few cells (8, 16, 24, and 32). Class 2 has strands of moderate memory depth (48, 64, 80, and 96) of typical 50 cells [48]. Class 3 has strands of enhanced memory depth (128, 160, 192, and 224 LSTM cells), used for moderate training tasks, while class 4 has LSTM strands of excessively long memory (288, 352, 416, and 480 LSTM cells), which are used for complex tasks of many attributes and increased time depths. Nevertheless, depending on the problem attributes and dataset, a vast number of cells may suffer from the vanishing gradient problem due to significant cell depths. Figure 1 illustrates the stranded models included in the proposed stranded LSTM model, as well as the selection and inference (final forecast output) steps.

Except for RNNs, the vanishing gradient issue may also occur in LSTMs (Long Short-Term Memory networks) when the number of cells (timesteps) or layers is high. While LSTMs were designed to mitigate vanishing gradients using gating mechanisms (input, forget, and output gates), they are still not immune to the problem. For example, if the gradient back-propagates through multiple LSTM cells and if the derivatives of the activation functions (sigmoid/tanh) are small (<1), or in cases of very long training sequences [49].

For this reason, an appropriate mathematical formula accommodates the calculation of the LSTM cells derived from experimentation using powers of 2 for computational efficiency, and a class hierarchy has been implemented to deal with the different types of time series micro–macro response and vanishing gradient issues in an automated unsupervised way. Using powers of two for LSTM depths creates an efficient hierarchy of temporal abstraction through the exponential coverage of the time domain, analogous to how musical octaves or wavelet transforms decompose signals [44,45].

As shown in Table 1, the use of class partitioning divides the time domain into bins, ensuring no timescale overlaps, minimal class redundancy, and uniform temporal spectrum coverage. Similarly, to the wavelet equivalent [44,45], in the stranded LSTM class hierarchy, class 1 corresponds to instantaneous or high-frequency detail patterns, class 2 corresponds to short-term or mid–high-frequency patterns, and class 3 corresponds to the acquisition of

mid–low-frequency temporal patterns. In addition, class 4 tries to capture low-frequency macroscopic trends.

In the proposed stranded LSTM model, class 1 initiates with an LSTM strand of 8 cells. This cell size is the most simple LSTM size used in the literature, providing predictions for small-size datasets or ultra-short timespans. Then, we add new LSTM strands to class 1, incrementing by step = $8(2^3)$ cells up to 32 cells, using as hyperparameter the class width or range factor (RF) calculated as follows: $RF = \frac{max_{depth}^{Class}}{min_{depth}^{Class}}$. The range factor (max depth/min depth per class) quantifies the relative breadth of temporal coverage within each LSTM class. The absolute span of each class expressed as $d_n^c - d_1^c$ is 24, 48, 96, and 192 for the four class model hierarchy. Therefore, it doubles as the class number increases to offer broad coverage of short timescales and long-term slight variations.

Focusing on stranded LSTM model classes, class 1 has been designed for high dynamic range tasks where both fine details (8 cell timesteps) and short-term trends (32 cell timesteps) matter. Above 32 cells, and using a fixed inter-class gap, the class 2 strands are incremented using a step of $16(2^4)$ cells. From 96 cells and above, class 3 resides using strands of $32(2^5)$ cell steps, and finally, from the 224 cell strand, class 4 is defined using a step size of $64(2^6)$ cells. In order to maintain uniformity (uniform plurality) in the contribution of the final output weights (per class selected strand), the number of strands per class is equally set in order to capture both long and short-term patterns. Therefore, class 1 was designed for high dynamic range data where fine details (8 cells) and short-term trends (32 cells) matter. Following a small class gap equal to the per-class LSTM strands cell step in that class, class 2 initiates. Class 2 has been designed for medium volatility signals, having a space of $2^4 = 16$ cells per strand. Then, class 3 initiates for medium granularity tasks of $2^5 = 32$ distances, and finally, class 4 focuses on macro-patterns, where individual steps are less critical and reside with a per-strand distance of $2^6 = 64$.

**Table 1.** Stranded LSTM classes' timescale coverage, computation intensity, and distinct temporal regimes. The per-class range factor that signifies each class width as well as inter-class gaps are considered hyperparameters for the model and can be arbitrarily set, as well as the per-class LSTM strand density, in order to satisfy the four-class hierarchy, as defined by timescale coverage and computation intensity. In this stranded LSTM model, inter-class gaps heuristically follow an 8, 16, 32 cell space logic and a 2, 0.25, 0.08 range factor reduction per class.

| Class | Depths | Timescale Coverage | Computation Intensity | Usage | Range Factor | Coverage |
|---|---|---|---|---|---|---|
| 1 | 8, 16, 24, 32 | Ultra short | Low (edge) | IoT sensors, audio signals | 4.0 | 4× wide timespan |
| | | | Inter class gap of $2^3 = 8$ cells | | | |
| 2 | 48, 64, 80, 96 | Short term | Medium (edge) | Video, stock data | 2.0 | 2× narrower than base |
| | | | Inter class gap of $2^4 = 16$ cells | | | |
| 3 | 128, 160, 192, 224 | Medium term | High | ECG | 1.75 | 25% narrower than class 2 |
| | | | Inter class gap of $2^5 = 32$ cells | | | |
| 4 | 288, 352, 416, 480 | Long term | Extreme | Genomics, climate | 1.67 | 8% narrower than class 3 |

All stranded LSTM model strands are trained over datasets that use one fixed time depth value, which is set as a training hyperparameter. This parameter is initially set based on the training scenario for all strands. This learning depth selection is accommodated by transforming the original time series dataset, where vector attributes are converted into continuous input chunks. Then, each chunk is annotated with the next vector value as the

forecast output. This data partitioning serves as a pre-training transformation of the raw dataset. The transformed data are then split into training, validation, and evaluation (test) sets based on the following ratios:

- For classification: 70–20–10% (training, validation, and evaluation).
- For forecasting: 70–10–10–10% (training, validation, evaluation, and testing).

The metrics described in Section 2.2 are computed for each strand during per-strand training and evaluation. These values are stored within the model as historical validation and evaluation data. Figure 2 illustrates the layered architecture of each strand, with representative strands from the four model classes.

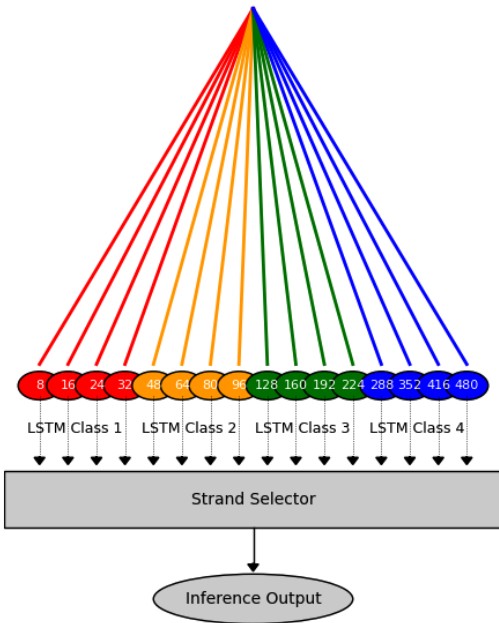

**Figure 1.** Proposed stranded LSTM model inference process providing measurement forecasts.

During the data training process, let a dataset of $k$ attributes (sensory values) be used for forecasting, and then the forecasting learning depth $d_{th}$ in sensor measurement batches needs to be specified. This depth value remains unchanged for all strands—only the number of cells per LSTM strand changes in the model. For forecasting cases, data inputs need to be transformed to (batches,input_size $= (d_{th}, k)$), where $k$ is the number of data attributes and $d_{th}$ is the data batch depth (or classification depth). The other important hyperparameter of the model is the forecast length $f_{len} = n_{of}\ forecasts \cdot k$. For classification problems, the forecast length equals the number of classes the model classifier implements.

Therefore, the labeled forecasting data include sensory data of ($f_{len}$,) and the classification ($n_c$,) of hot encoded annotated class outputs. Then, two fully connected NN layers follow, having $2 \cdot f_{len}$ number of neurons. For Figure 2, the number of forecasts (forecast batch values) equals 6, and for $k = 3$, the NN layers are comprised of two 36 fully connected neurons with tanh and linear or ReLU activation functions accordingly. Then, the outputs of the NN layer are reshaped to a 2D matrix of ($n = 2 \cdot n_{of}\ forecasts$, $k$). In this reshaped array, a batch normalization process applies. Then, it is flattened again to a $2 \cdot f_{len}$ NN layer followed by a $f_{len}$ NN layer, both fully connected with ReLU (1st layer) and tanh (2nd layer) activation functions accordingly. This final layer output of $f_{len}$ values (of classification outputs if classification is performed) corresponds to the per-strand output vector of values. For forecasting cases, this corresponds to $n_{of}\ forecasts \cdot k$ attributes. Figure 2 illustrates the generic model construction process for each LSTM strand.

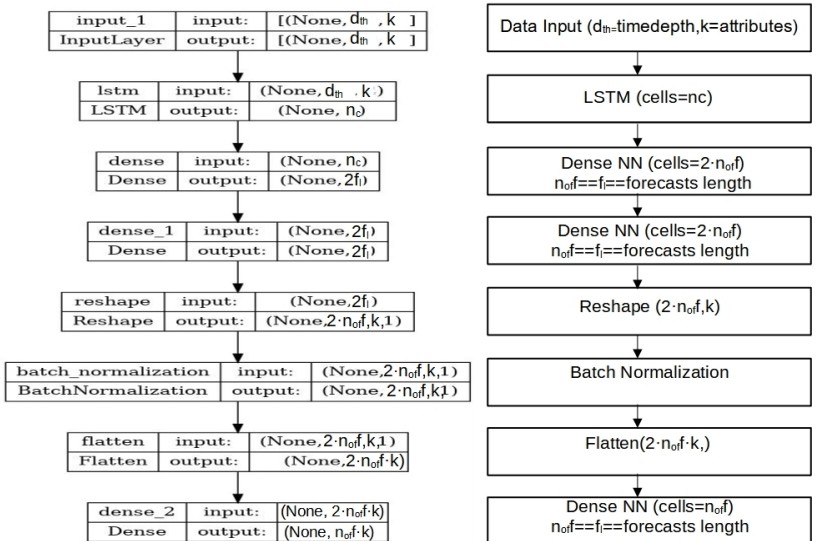

**Figure 2.** Generic construction flow diagram of each LSTM strand having $n_c$ as the number of cells, $n_{of}f = n_{of}$ $forecasts$, the number of forecast steps, $k$ as the number of attributes, $d_{th}$ as the learning depth, and $f_l = f_{len} = n_{of}$ $forecasts \cdot k$.

The model's multiple training processes per class and inference process are illustrated in Figure 3. For Figure 3, data input is set to ($d_t h = 5$, $k = 3$). Then, for each LSTM strand, a different cell depth strand trains on the data, providing several values by the LSTM cell length as output. The output of 18 values per strand corresponds to 6 forecasts of the three data attributes. The inference process follows a similar logic. Let input inference data for forecasting of ($d_{th}$, $k$), where $d_{th}$ the history depth and $k$ the data attributes. Then, according to the model-selected strategy described in Section 2.3, a strand or multiple LSTM strands will be selected to provide inferences, and according to the selected strategy weights, the weighted aggregation of LSTM strands inferences will be the model's forecasting output of forecasting length $f_{len}$. For classification problems, the classification vector ($c_v$, 1) is used instead of the ($d_{th}$, $k$) array.

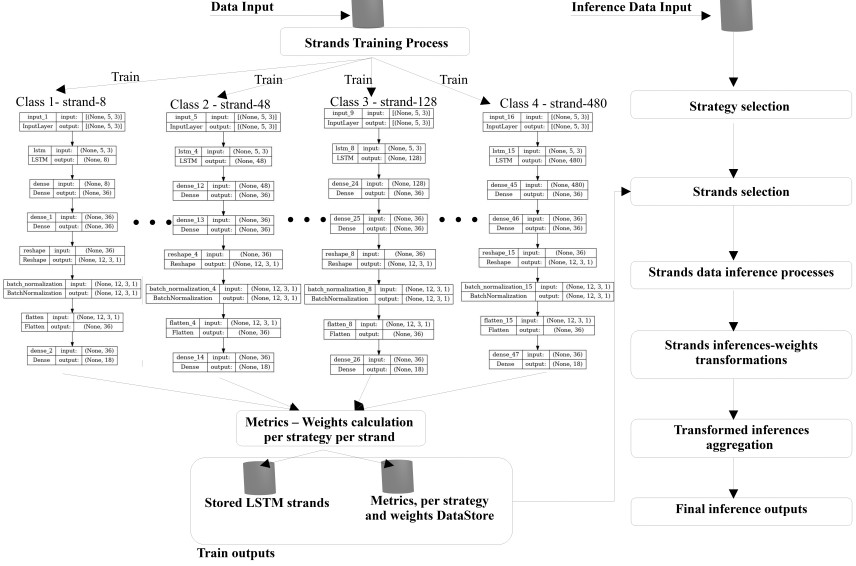

**Figure 3.** Internal design of the LSTM strands and flow diagram of the training and inference processes.

Section 2.2 outlines the primary metrics used, including the Mean Squared Error (*MSE*) and jitter, which the stranded LSTM selection strategies employ for weight calculation during training. Section 2.3 then provides a detailed description of the supported model inference strategy selectors.

### 2.2. Metrics Used

In order for a strand selection process to apply, called as a strategy to evaluate the LSTM strands for the generation of the forecasting outputs, the following metrics are used during training, evaluation, and inference processes. The Mean Squared Error (*MSE*) metric is the base metric used during the training and validation steps of each one of the LSTM strands. The *MSE* loss function is commonly used in regression tasks to measure the average squared difference between predicted and actual values. When dealing with multiple sensory measurements, the *MSE* can be extended to handle multiple outputs (e.g., numerous sensor values of temperature and humidity taken from different locations inside the measurement area). Let $i = 1\ldots n$ be the number of past time measurements (measurements time depth), $j = 1\ldots m$ the number of sensory output values (input sensory attributes), $y_{ij}$ the actual sensory measurement, and $\hat{y}_{ij}$ the predicted $L_{MSE}$ loss formula is calculated using Equation (1).

$$L_{MSE} = \frac{1}{n \cdot m} \sum_{i=1}^{n} \sum_{j=1}^{m} (y_{ij} - \hat{y}_{ij})^2 \tag{1}$$

The Mean Squared Error is the primary metric used for validating and evaluating model training, especially in the context of measurement forecasts. However, other metrics such as the Root Mean Squared Error (*RMSE*) or Mean Absolute Error (*MAE*) can also be employed in place of the *MSE* [50]. Additionally, when selecting the best Long Short-Term Memory (LSTM) model for forecasting, it is important to consider the loss jitter metric alongside the chosen loss function.

Jitter refers to the variation in packet delay over time and is commonly analyzed in computer networks. It is calculated as the mean deviation of successive packet delay variations, inter-arrival packet differences, or inter-frame gaps [51]. In the context of *MSE* loss, jitter indicates the fluctuations in *MSE* loss values over time. This specific metric is referred to as *MSE* jitter, and it is computed according to Equation (2), which measures the differences between two consecutive *MSE* loss values. The negative sign in this calculation represents the expected decrease in *MSE* validation values during training, indicating that $MSE_l \geq MSE_{l+1}$.

$$J_{MSE_l} = -(MSE_{l+1} - MSE_l) \tag{2}$$

where $l = 1\ldots k$ are *MSE* losses of arbitrary time intervals (time data chunks). This jitter metric does not try to identify fluctuations between measurement time frames but to examine potential abnormalities during strand training and evaluation, expressed by inference variability. Therefore, the positive or negative signs are not taken into account. Jitter values are further distinguished into validation and evaluation jitter. Evaluation jitter pinpoints the final loss variation of the model over an unknown dataset, while the standard deviation of validation jitter ($\Delta MSE$), as calculated by Equation (3), gives a rough estimate of the *MSE* jitter over epochs during training.

$$\bar{J}_{val} = \sqrt{\frac{1}{N-1} \sum_{k=1}^{N-1} (J_{MSE_k} - \mu_{J_{MSE}})^2} \tag{3}$$

where $k = 1 \ldots N$ represents the training iterations in an epoch, $J_{MSE_k}$ represents the iteration $k$ loss, and $\mu_{J_{MSE}}$ is the mean loss for all $k$ in an epoch calculated as follows: $\mu_{J_{MSE}} = \frac{1}{N} \sum_{k=1}^{N} J_{MSE_k}$. Similarly, the evaluation jitter is calculated according to Equation (3).

If the $MSE$ jitter values are high, it indicates that the training process has considerable variability. This variability leads to less confident estimates and increased noise, resulting in a higher estimation of gradient noise. The boundary between stability and instability varies depending on the scale of the $MSE$ loss, the characteristics of the dataset, and the specific problem domain. However, for the sensory measurement case studies discussed in this paper, we can provide general guidelines based on observations related to the calculations of evaluation jitter. This differentiation further assists in clarifying among instances of overfitting or underfitting. The following three distinct cases are of particular interest:

Low jitter cases ($J_{MSE} < 0.01$): This scenario is characterized by minimal fluctuations in loss, indicating that the model is learning or forecasting smoothly without erratic variations.

Moderate jitter cases ($0.01 \leq J_{MSE} < 0.07$): In this case, some fluctuations in loss are present, but they do not substantially affect the training process. This level of jitter is typical for larger datasets with shuffled data batches. However, moderate jitter during the evaluation phase may signify either the presence of noise in the dataset or initial signs of a model that is not converging adequately.

High jitter cases ($J_{MSE} \geq 0.07$): This scenario suggests a highly unstable model. If the validation jitter is low while the evaluation jitter is high, it strongly indicates overfitting. Should the difference between these values exceed 0.1, it points to an underfitting issue. $MSE$ evaluation values exceeding 0.07 in most measurement forecasting contexts indicate poor model convergence and instability.

The combination of both the validation jitter and evaluation jitter values is used by the strand selector of the stranded LSTM model to provide a uniform metric of total jitter, expressed through an Exponentially Weighted Moving Average (EWMA) process, as presented in Equation (4).

$$J_{tot} = \alpha \cdot \bar{J}_{val} + (1 - \alpha) \cdot J_{eval} \tag{4}$$

where $\bar{J}_{val}$ is the mean validation jitter over epochs calculated by Equation (3) and $J_{eval}$ is the evaluation jitter calculated by Equation (2). Parameter $\alpha$ is a weighting factor controlling the balance between validation jitter and evaluation jitter. ($0 < \alpha \leq 1$). A small value of $\alpha (\alpha \leq 0.2)$ gives more importance to the evaluation jitter value, while a value above 0.7 gives more importance to the training validation jitter.

Following the calculation of the LSTM strands' evaluation loss using Equation (1) and the computation of the total jitter loss for each strand (using Equation (4)), the LSTM selection process begins (see Figure 1). This process involves (1) selecting the best representative candidate for each strand class and (2) determining the weights for the class representative strands. These weights are essential for generating the final array of forecast attribute values for a given time interval $k$, as described in Section 2.3.

### 2.3. Stranded LSTM Selection Process

The stranded LSTM selection process is automatically initiated at the end of the evaluation process and is performed to select the best candidate strand per class. At first, a standardization process using Z-score normalization or standardization (standardization

$S = \frac{x-\bar{x}}{\sigma}$) for all strand means loss values are conducted, assuming a normal loss distribution of the mean of 0 and a standard deviation of 1 (see Equation (5)), followed by the normalization of total jitter values according to min–max normalization using Equation (6).

$$||L_i|| = \frac{L_i - \bar{L}}{\sigma} \tag{5}$$

where $\bar{L} = \frac{1}{N}\sum_{i=1}^{N} L_i$ is the mean loss and $\sigma = \sqrt{\frac{\sum_{i=1}^{n}(L_i-\bar{L})^2}{n-1}}$ is the sample standard deviation of the $i = 1..N$ model LSTM strands.

$$||\bar{J}_i|| = \frac{\bar{J}_i - \bar{J}_{min}}{\bar{J}_{max} - \bar{J}_{min}} \tag{6}$$

The $J_i$ is the total jitter loss per strand $i$ as expressed by Equation (4) and $i = 1..N$ are the stranded LSTM model strands. The final normalization output consists of two vectors $\vec{L}$ and $\vec{J}$ that include as elements the normalized $||L_i||$ and $||\bar{J}_i||$ values accordingly. Let $\vec{L}$ and $\vec{J}$ be the metric vectors with $n$ normalized elements each. We can represent them as follows:

$$\vec{L} = (||L_1||, ||L_2||, \ldots, ||L_n||)$$

and

$$\vec{J} = (||J_1||, ||J_2||, \ldots, ||J_n||)$$

where each $||L_i||$ and $||J_i||$ values have an associated class $c_j \in \{1, 2, 3, 4\}$. To find the element within each class $c$ that minimizes both $L$ and $J$ (assuming we want to minimize them simultaneously), we can define the following sets for each class:

$$I_c = \{i \in \{1, 2, \ldots, n\} \mid c_i = c\}, \quad \text{for } c = 1, 2, 3, 4$$

These $I_c$ elements contain the indices of the strands that belong to a class $c$. Then, for each class $c$, we can find the strands that minimize both $||L_i||$ and $||J_i||$ within that class set by minimizing the sum according to Equation (7):

$$i_c^* = \arg\min_{i \in I_c}(||L_i|| + ||J_i||) \tag{7}$$

The approach expressed by Equation 7 may shadow the jitter metric value due to the difference in magnitude between loss and jitter. A big jitter value will not be punished accordingly to a big loss value. Therefore, Equation (7) is transformed to Equation (8) using multiplication instead of addition to express the appearance of two independent events and replacing $||J_i||$ with the natural logarithm as $ln(1 + ||J_i||)$ to increase the importance magnitude of the loss variance as used similarly by to the cross-entropy loss metric [52].

$$i_c^* = \arg\min_{i \in I_c}(||L_i|| \cdot \ln(K_p + ||J_i||)) \xrightarrow{K_p=1} i_c^* = \arg\min_{i \in I_c}(||L_i|| \cdot \ln(1 + ||J_i||)) \tag{8}$$

where $K_p \geq 1$ is a coefficient with a default value of 1. A $K_p$ value greater than 1 can further increase the importance of jitter over loss. Furthermore, validation jitter is affected by poorly chosen momentum hyperparameter values, particularly when they are set too high. During back-propagation, adaptive learning rates may lead to overshooting, which causes instability. However, this effect impacts all LSTM strands since the learning rate is shared across all strands during training. Consequently, the kp parameter in Equation (8) is set to a minimum value of 1. Therefore, expression $\ln(Kp + ||J_i||)$ has an upper boundary value of 10 for very big jitter values $||J_i|| \leq 20,000$. This adjustment assists in mitigating erroneous jitter values by effectively increasing the significance of evaluation loss compared to jitter

for the weight calculation process. Nonetheless, if the learning rate experiences spikes due to an adaptive learning rate policy, it will affect validation losses and, consequently, all models. Since all models are trained sequentially on the same dataset, these effects can uniformly propagate throughout all LSTM strands.

This selective process, described by Equation (9), leads to the selection of the best representative $i$ strand metric values of loss $||L_i||$ and total jitter $||J_i||$ that corresponds to a class $c = 1 \dots 4$. The final output for all classes is expressed using an output vector $\vec{u} = (i_1^*, i_2^*, i_3^*, i_4^*)$. Having as input the $\vec{u}$ and the selected loss and jitter strand values per class, three different strategies may apply for the calculation of the final forecasting output results, as follows:

**Least loss selection strategy:** This is a classless strategy, and as the name implies, the LSTM strand $i$ that has the minimum value of $||L_i|| \cdot \ln(K_p + ||J_i||$ across all model strands, as expressed by Equation (9).

$$i^* = \arg\min_{i \in I_n}(||L_i|| \cdot \ln(K_p + ||J_i||)) \xrightarrow{K_p=1} i^* = \arg\min_{i \in I_n}(||L_i|| \cdot \ln(1 + ||J_i||)) \tag{9}$$

where $I_n$ is not $I_c$ set of class strands (as expressed by Equation 8) but the set of all strands. The coefficient $K_p \geq 1$ has a default value of 1, signifying jitter over loss importance. The selected $i^*$ strand is explicitly used to calculate the output inference values. This strategy is a minimalistic process that ignores that the model uses classes of different cell depths to provide more selective forecasts. This strategy requires dense training and evaluation periods when new data are added to the dataset to maintain minimal prediction losses; nevertheless, training datasets of fixed size trained once may provide good accuracy results, specifically for small datasets.

**Weighted least loss selection strategy:** This strategy takes into account all the representative strands of the four classes as calculated in the $\vec{u}$ using Equation (8), as well as their corresponding loss-jitter values expressed as follows: $|I_{i_{c_j}}| = ||L_{i_{c_j}}|| \cdot log(K_p + ||J_{i_{c_j}}||), k_p = 1$. That is, after calculating vector weight indices where each vector corresponds to per-class selected output strand, weighted inferences using the selected class vectors $\vec{u}$ as multiplicative terms are aggregated to provide the final inference result of a classification or a forecast scenario.

Let $i^*$ be the number of the selected element in each class $c$ of the set $I_c = (i_1^*, i_2^*, i_3^*, i_4^*)$, and its corresponding loss-jitter value is denoted as $|I_{i_{c_j}}|$. The total sum of the selected elements jitter loss values $s = \sum_{c=1}^{4} |I_c|_i = \sum_{i=1}^{4} |I_{i_{c_j}}|, i \in c \text{ and } j \in i^*)$. The normalized weight $w_c$ of each class element in $c$ is calculated using Equation (10)

$$w_{c_j} = 1 - \frac{I_{c_j} - I_{min}}{I_{max} - I_{min}}, \quad \text{for } c = 1, 2, 3, 4 \tag{10}$$

where $I_{min} = min\{I_j\}, I_{max} = max\{I_j\}$, where $j = 1..N$ the number of model strands. These $w_{c_j}$ weights represent each class's relative importance or prevalence to the other classes. They also satisfy the condition that they sum up to 1, meaning that the equivalent weight of $w'_{c_j}$ of a class c=1,2,3,4 is calculated as follows: $w'_{c_j} = \frac{w_{c_j}}{\sum_{c=1}^{4} w_{c_j}}$, meaning that $\sum_{c=1}^{4} w'_{c_j} = 1$.

Let $T$ be the $n \times m$ matrix representing the $n$ time forecast values of $m$ attributes from a selected class strand. We can denote the elements of $T$ as $T_{ij}$, where $i = 1, 2, \dots, n$ represents the time index, and $j = 1, 2, \dots, m$ represents the attribute index.

Let $W$ be a $1 \times 4$ weight matrix. We can denote the elements of $W$ as $W_c = w_{c_j}$, where $c = 1, 2, 3, 4$ represents the class index and $j \in I_c$ represents the selected strand index. In this strategy, each element of $T$ is multiplied by each $W$ matrix element. This results in four

separate $n \times m$ matrices. Let us call these resulting matrices $R^{(c)}$, where $c = 1, 2, 3, 4$. It can be represented in matrix notation using the Equation (11) element-wise product denoted by $\odot$ symbol.

$$R_{(c)} = W_{(c)} \odot T_{(c)}, \quad \text{for } c = 1, 2, 3, 4 \tag{11}$$

where $W^{(c)}$ is the weight value for class c as calculated by Equation (10), and $T^{(c)}$ are the class c predicted. Finally, $R^{(c)} = (R^{(i=1)}, ..., R^{(i=c)})$ are the per class weighted min–max normalized inference responses. So, for each class $c$, we are scaling its entire forecasts matrix $T$ by the corresponding normalized weight $w_{c_j}$. The final forecast array $R'$ is calculated by adding all $R^{(c)}$ elements according to Equation (12).

$$R' = \sum_{c=1}^{4} R^{(c)} \tag{12}$$

This selection strategy involves choosing at least one strand from various classes, including LSTM models with different cell depths. Incorporating strands from multiple depth classes helps mitigate the vanishing gradient issue, if it arises, and allows for a more comprehensive understanding of the best learners within each class. However, this strategy may also introduce inaccuracies due to less precise strands contributing to the final output. Unlike the least-loss selection strategy, this approach does not require frequent re-evaluation periods. Consequently, it can perform better on large datasets, providing more accurate inferences compared to the least element selection process.

**Fuzzy least loss selection strategy:** In this fuzzy selection process, instead of directly using loss and jitter loss values to calculate weights, fuzzy sets are defined for them. At first, for each class c, the total jitter loss $J_{tot}$ is computed based on Equations (3) and (4). Then, fuzzy sets are mapped into each class's loss $L_c$ and total jitter loss $J_{c_{tot}}$. Three fuzzy sets of high, medium, and low loss are used for loss and jitter loss accordingly. Then, for all classes and strands, the $L_m in$, $L_m ax$, $J_m in$, and $J_m ax$ values are calculated, and for each class, the mean class strand loss $L_c = \bar{L}_c$ and total jitter loss $\bar{J}_c$ are calculated accordingly. The membership functions that define the degree $\mu_L$ of belonging to low, medium, and high loss sets are based on the following set of Equations (13).

$$
\mu_L^{low} = \begin{cases} 1, & L_c \leq L_{min} \\ \frac{L_{max} - L_c}{L_{max} - L_{min}}, & L_{min} < L_c < L_{max} \\ 0, & L_c \geq L_{max} \end{cases}
$$

$$
\mu_L^{med} = \begin{cases} 0, & L_c \leq L_{min} \text{ or } L_c \geq L_{max} \\ \frac{L_c - L_{min}}{\frac{L_{max} - L_{min}}{2}}, & L_{min} < L_c \leq \frac{L_{max} + L_{min}}{2} \\ \frac{L_{max} - L_c}{\frac{L_{max} - L_{min}}{2}}, & \frac{L_{max} + L_{min}}{2} < L_c < L_{max} \end{cases} \tag{13}
$$

$$
\mu_L^{high} = \begin{cases} 0, & L_c \leq L_{min} \\ \frac{L_c - L_{min}}{L_{max} - L_{min}}, & L_{min} < L_c < L_{max} \\ 1, & L_c \geq L_{max} \end{cases}
$$

The medium membership function has a triangular shape peaking at $L_c = \frac{L_{max} + L_{min}}{2}$, ensuring a smooth transition between fuzzy sets. Similarly, this applies for total jitter $J_c$ according to Equations (14).

$$\mu_J^{low} = \begin{cases} 1, & J_c \le J_{min} \\ \frac{J_{max} - J_c}{J_{max} - J_{min}}, & J_{min} < J_c < J_{max} \\ 0, & J_c \ge J_{max} \end{cases}$$

$$\mu_J^{med} = \begin{cases} 0, & J_c \le J_{min} \text{ or } J_c \ge J_{max} \\ \frac{J_c - J_{min}}{\frac{J_{max} - J_{min}}{2}}, & J_{min} < J_c \le \frac{J_{max} + J_{min}}{2} \\ \frac{J_{max} - J_c}{\frac{J_{max} - J_{min}}{2}}, & \frac{J_{max} + J_{min}}{2} < J_c < J_{max} \end{cases} \tag{14}$$

$$\mu_J^{high} = \begin{cases} 0, & J_c \le J_{min} \\ \frac{J_c - J_{min}}{J_{max} - J_{min}}, & J_{min} < J_c < J_{max} \\ 1, & J_c \ge J_{max} \end{cases}$$

The medium membership function to total jitter is also a triangular function centered at $J_c = \frac{J_{min} + J_{max}}{2}$. The fuzzy weight probabilistic metric for each class c ($FW_c$) is defined according to Equation (15).

$$FW_c = \frac{\mu_{L,c}^{low} + \mu_{J,c}^{low}}{\mu_{L,c}^{low} + \mu_{J,c}^{low} + \mu_{L,c}^{med} + \mu_{J,c}^{med} + \mu_{L,c}^{high} + \mu_{J,c}^{high}} \tag{15}$$

where $mu_{L,c}^{low}, \mu_{L,c}^{med}, \mu_{L,c}^{high}$ are the fuzzy membership degrees of loss for class c and $mu_{J,c}^{low}, \mu_{J,c}^{med}, \mu_{J,c}^{high}$ the corresponding fuzzy membership degrees of jitter for class c. Equation 15 ensures that model classes with lower loss and total jitter values receive higher $FW_c$ weight values (closer to 1), while model classes with higher loss and total jitter values obtain lower $FW_c$ values (close to 0). Finally, given four forecast matrices $T_c$, with one strand representative for each class selected from the per class strand that maintains a maximum min–max normalized weight value—$i_c^* = \arg \max_{i \in I_c} \frac{w_i - w_{min}}{w_{max} - w_{min}}$—that is also inferred in forecasted timesteps and m attributes—$T_c \in \mathbb{R}^{n \times m}, c = 1, 2, 3, 4$. The final forecast values are calculated according to Equation (16).

$$FW_c = \sum_{c=1}^{4} FW_c \odot T_c \tag{16}$$

Using a fuzzy strategy provides a smoother handling of uncertainty than hard threshold values and a more robust weight calculation based on fuzzy logic. This strategy applies smooth fuzzy transitions between loss and jitter handled by fuzzy probabilities rather than a deterministic equation described by the weighted inferences aggregation process of Equation (9), as proposed by the weighted balanced selection strategy.

Based on the aggregated inference strategy used—either weighted-based or fuzzy—and the previously provided WPM calculation formulas, each selected element (c = 1..4) is assigned a WPM value that corresponds to its loss and total jitter values. Table 2 illustrates the trends in evaluation loss, jitter, and WPM values, as calculated using both the weighted-based and fuzzy strategies for each selected class element. These WPM values are then sum-normalized to produce the final $w'$ WPM values. These values are used to provide a weighted sum of class-strands model inferences that is the final output for both forecasting and classification cases.

**Table 2.** LSTM strand condition based on the strand selector process metrics described in Section 2.2 and the WPM equations for weighted and fuzzy strategies.

| Condition | MSE Loss | Total Jitter | WPM $w'$ Value | Meaning |
|---|---|---|---|---|
| Best Case (Ideal) | Close to 0 | Close to 0 | Close to 1 | Excellent classification or forecasting |
| Worst Case (Loss and Jitter) | Close to 1 | Close to 1 | Close to 0 | Bad classifier or predictor [1] |

[1] due to overfitting or underfitting

The following Section 3 presents the authors' experimentation of their proposed stranded LSTM model and comparison results to the LSTM model.

## 3. Experimental Scenarios and Results

Two experimental scenarios have been used to evaluate the proposed stranded LSTM implementation of two different datasets. The first scenario involves classifying the temperature of an industrial compressor [13]. The second scenario focuses on forecasting temperature and humidity based on measurements taken from IoT sensors in the thingsAI system, which are employed in viticulture to detect downy mildew [11]. Section 3.1 details the data classification scenario, while Section 3.2 describes the viticulture forecasting scenario. The aforementioned datasets were real-case scenario datasets maintained by the authors and were used to compare their stranded LSTM implementations for classification problems with existing LSTM model results. The *MSE* losses from the actual values were used for the forecasting scenario.

### 3.1. Classification Scenario

The classification scenario involves data inputs from an industrial oil industry compressor. The dataset contains 16 temperature measurements collected from various points in the first stage cylinders N.3 (SET 3, with eight sensors) and N.4 (SET 7, also with eight sensors). The dataset includes records of 16 temperature readings per minute, which are associated with alerts for mean SET temperatures exceeding 60 °C [13]. This dataset has a total of 7,439,584 temperature samples, representing 11 months of minute-by-minute temperature measurements. Each temperature sample has been annotated using a threshold-based temperature annotation process, as outlined in Table 3.

**Table 3.** Annotation process of a given set of temperature values $T_i$, related to a given *alert* threshold (temperatures above *alert* = 60 °C).

| Class | State | Range of Temperature $T_i$ |
|---|---|---|
| 0 | Normal | $0 \leq T_i < 0.25 * alert$ |
| 1 | Low-Risk | $0.25 * alert \leq T_i < 0.5 * alert$ |
| 2 | Caution | $0.5 * alert \leq T_i < 0.75 * alert$ |
| 3 | Critical | $0.75 * alert \leq T_i < alert$ |
| 4 | Danger | $T_i \geq alert$ |

Each dataset record consists of 16 temperature values $T = \{t_1, t_2, \ldots, t_{16}\}$ measured per minute, and each temperature measurement $T_l$ has an associated class value $C_l$, as indicated in Table 3, where $C_l = F(T_l)$ and $C_l \in \{0, 1, 2, 3, 4\}$. To determine the class value for the entire set, we compute the average as follows:

$$\bar{C} = \frac{1}{16} \sum_{l=1}^{16} C_l.$$

Before model training, the dataset records have been split to 70% for training, 20% for validation, and 10% for evaluation. Then, each stranded model was trained, and the selected strand or weight matrices were calculated depending on the strategy used (least loss, weighted least loss, and fuzzy least loss). The model strands training hyperparameters include a batch size of 32, the Adam optimizer [53] with a small learning rate of 0.001, categorical cross-entropy as the loss function [52], a total of 50 training epochs.

Maintaining the same weights for each LSTM strand, the testing phase was initiated by performing classification predictions. The output from each strand's LSTM model is a one-hot encoded vector of probabilities with $m$ elements corresponding to the classes 0 through 4 (i.e., $v_0, v_1, \ldots, v_4$). The predicted class position is determined using the softmax function for classification:

$$i^*_{softmax} = \arg\max_i \sigma(v_i),$$

where

$$\sigma(v_i) = \frac{e^{v_i}}{\sum_{j=0}^{4} e^{v_j}}.$$

By using the selected strand for the least loss inferences and the weighted aggregated inferences for the weighted and fuzzy strategies, the final inference outputs per strategy per input batch were calculated and then passed to a softmax function to select the best candidate predicted class for each batch input of the test set. Then, the mean accuracy and loss metrics were calculated and compared to the actual values. The whole experiment was repeated for 10 cycles, following a random batch split of the testing dataset and the mean metric values and weights calculation.

Table 4 presents the model results using the three different selection strategies for testing the accuracy and cross-entropy loss. It also presents the evaluation results of two independent LSTM models that consist of two arbitrary LSTM layers with an equal number of cell units ($nc$). The first is an LSTM model with $n_c = 16$ cells per layer, while the latter is an LSTM model with $n_c = 60$ cells per layer. The reason dual-layer LSTMs were used is that, as mentioned in [13], dual-layer LSTMs outperformed single LSTMs of the same cell length.

**Table 4.** Cross comparison of the stranded LSTM model's different strategies to dual-layer stacked LSTM networks of small and medium cell numbers $n_c = 2 \times 16$ and $n_c = 2 \times 60$.

| Model | Timestep ($d_{th}$) | Selected Strand | | | | Results | |
| --- | --- | --- | --- | --- | --- | --- | --- |
| | | C1 | C2 | C3 | C4 | Loss | Accuracy |
| LSTM $2 \times 16$ | 200 | - | - | - | - | 0.116 | 0.952 |
| LSTM $2 \times 60$ | 200 | - | - | - | - | 0.194 | 0.910 |
| Stranded LSTM least loss selection | 200 | 32 | 32 | 32 | 32 | 0.082 | 0.961 |
| Stranded LSTM weighted least loss selection | 200 | 32 | 48 | 128 | 288 | 0.075 | 0.967 |
| Stranded LSTM fuzzy least loss selection | 200 | 32 | 48 | 128 | 288 | 0.053 | 0.974 |

The experimental results indicate that the stranded LSTM models outperform the best candidate among the LSTM models, specifically the $2 \times 16$ LSTM configuration, by 0.9% to 2.3% in terms of accuracy. Among the three strategies supported by the stranded LSTM model, the least loss selection process yields the lowest accuracy. This process selects the

$n_c = 32$ cells strand of class 1 as the model's representative LSTM network, achieving an accuracy of 96.1%. This result is 0.9% better than the best dual-layer $2 \times 16$ LSTM model and 5.6% better than the $2 \times 60$ LSTM model.

The stranded LSTM model utilizing the weighted least loss selection strategy achieves an accuracy that is 0.6% higher than that of the stranded LSTM least selection model. Furthermore, the fuzzy least loss selection strategy surpasses both previous strategies by 1.3% and 0.7%, respectively. Notably, the weighted and fuzzy least loss strategies derive similar representative strands for the same classes, with nearly identical weight values for classes 3 and 4. However, the fuzzy least loss selection process assigns a slightly higher weight index value (0.07 more) for class 32 than the weighted least loss selection process.

In conclusion, all three strategies outperform the $2 \times 16$ LSTM model by an average of 1.5% (with a maximum improvement of 2.3% for the fuzzy model) and the $2 \times 60$ LSTM model by an average of 6.4% (with a maximum improvement of 7% for the fuzzy model). The next section will describe a forecasting scenario using the stranded LSTM model and present the results.

### 3.2. Forecasting Scenario

The forecasting scenario dataset consists of sensory measurements collected from a meteorological sensor within the thingsAI system, which is used for micro-climate monitoring in viticulture. This system focuses on detecting diseases, specifically providing intervention suggestions and alerts for downy mildew. The dataset includes the following two attributes: temperature and humidity measurements collected from a single measurement point every minute. These measurements span from 2010 to 2023, covering the months from March to September, which corresponds to the vine cultivation and growth period leading up to harvest. The dataset comprises 3,709,440 pairs of temperature and humidity measurements, amounting to 7,418,880 individual measurements during this time frame.

The model training process was conducted using a depth of $d_{th} = 240$, which corresponds to four hours of measurements taken every minute for $k = 2$ attributes as follows: temperature and humidity. These depth numbers are indicative for nowcasting and micro-climate modeling [54,55]. These attributes were min–max scaled to a range of 0 to 1 to yield small $MSE$ (Mean Squared Error—square of the $RMSE$) loss values, comparable to those produced in a classification scenario using cross-entropy loss. The forecasting outputs comprise 240 future measurements (in minutes) for the two attributes (temperature and humidity). That is, the dataset has been transformed into a one-dimensional array of measurements with the dimensions $240 \times 2$ for each data label.

The model training hyperparameters include a batch size of 32, the Adam optimizer with a small learning rate of 0.001, and $MSE$ loss as the loss function. The model was trained for 50 epochs, and the dataset was split as follows: 70% for training, 10% for validation, 10% for evaluation, and 10% for testing, which was used for calculating the predictions and their corresponding $MSE$ losses. Table 5 presents the model results for forecasting using three different selection strategies and their associated $MSE$ losses.

The results presented in Table 5 show that the least loss strategy had the highest loss value, followed closely by the weighted least loss strategy, which achieved a loss that was 5% lower. The fuzzy least-loss selection strategy outperformed both least-loss and weighted strategies, delivering a loss that was 2.5% lower than the weighted least loss and 7% lower than the least loss. Notably, the least-loss selector's strongest performer was the LSTM model, with the maximum number of cells (480). This implies that aggregation inference policies can reduce loss compared to LSTM models by at least 5%. Moreover, fuzzy-based aggregation policies can achieve even more significant reductions, resulting in up to 7% less $MSE$ loss than individual LSTM models. The effectiveness of the fuzzy

strategies is evident, as they align with the trend of reduced losses associated with models containing more LSTM cells.

**Table 5.** Cross comparison of the stranded LSTM model different strategies in a forecasting micro-climate monitoring scenario using temperature and humidity measurements as input and *MSE* loss of temperature–humidity predictions to actual values.

| Model | Timestep ($d_{th}$) | Selected Strand | | | | Results Loss |
| --- | --- | --- | --- | --- | --- | --- |
| | | C1 | C2 | C3 | C4 | |
| | | $w_c$ | $w_c$ | $w_c$ | $w_c$ | |
| Stranded LSTM least loss selection | 240 | 480 | 480 | 480 | 480 | 0.075 |
| | | - | - | - | - | |
| Stranded LSTM weighted least loss selection | 240 | 32 | 96 | 192 | 416 | 0.071 |
| | | 0.04 | 0.047 | 0.24 | 0.673 | |
| Stranded LSTM fuzzy least loss selection | 240 | 32 | 96 | 224 | 480 | 0.0697 |
| | | 0.02 | 0.032 | 0.15 | 0.798 | |

In the results from classes 3 and 4, the weighted least loss selection policy struggled to adhere to this trend, as it selected previous strands with normalized weight differences of less than 0.05 value. In contrast, the fuzzy selector successfully identified the appropriate strands with the maximum number of cells in these classes. Section 4 summarizes the evaluation findings for stranded LSTM from the experiments conducted.

## 4. Discussion of the Results

The authors introduce a new model called stranded LSTM, which incorporates multiple LSTM models as a continuation of their previous stranded models in the literature [11,13]. This model utilizes LSTM cells rather than multiple layers of neurons and is designed to include a batch normalization layer within each strand. This addition helps to normalize activations, addressing issues such as internal covariate shift [56] and vanishing gradients [49,57]. Consequently, the proposed model at the strand level is not a pure LSTM model but rather a hybrid LSTM-NN model with a normalization layer.

The strength of the stranded LSTM model lies in its use of multiple LSTM models, each containing variable cells. During training, these models can either be directed to provide inferences individually or automatically select the strand that demonstrates the best performance, minimizing loss and jitter during evaluation and validation. This "least loss scheduling" strategy results in higher accuracy and robustness compared to a manually selected LSTM model. This capability is particularly valuable in training environments where models continuously learn from new data using either supervised or unsupervised learning processes supported by autoencoders. However, a drawback of this approach is that inference times increase in proportion to the number of strands utilized by the model.

To mitigate the negative effects of excessive training durations, our model incorporates a hierarchical structure along with two additional strategies: The weighted and fuzzy least loss strategies. This framework operates within a four-class hierarchy that clearly indicates the lengths of time required to memorize patterns, which is expressed through the number of LSTM strand cells in each class. Consequently, we have introduced the following four memory depth categories: low, moderate, enhanced, and excessive. This classification is based on an easy-to-implement variation of cells that follows powers of two.

The first step in this process is to select a representative strand for each class. Following this, we automate the removal of less important strands in each class after several training

iterations. Moreover, these strategies enhance the ability of the stranded LSTM model to perform ensemble learning by providing aggregative inferences through either bootstrap weight-based or fuzzy-based inference aggregation. However, a significant drawback of this model is that inference times are increased by at least four times compared to the time required for a single LSTM strand or a standard LSTM network.

The experimentation with the stranded LSTM model as a classifier revealed that it not only identified the best LSTM candidate for the classification task but also outperformed dual-layered LSTM models manually assigned for that task by at least 0.9–1%. Additionally, when employing aggregation strategies, it exceeded the performance of these dual-layered models by approximately 1.5–2.5%. Ultimately, these aggregation strategies outperformed forecasts from single LSTM models selected based on the least loss strand selection strategy, reducing *MSE* losses by at least 2–5%, with some cases reaching up to 7%.

The stranded LSTM model forecasting results in Table 5 illustrate *MSE* test loss ranges between 0.0697 and 0.075, with a mean *MSE* value of 0.072. Compared to the results presented at [54], the best single-layer LSTM regressor of 100 cells provided *MSE* loss values of around 6.63 for micro-climate rainfall forecasting. In addition, the authors at [58] experimented with dual-layered BiLSTMs of 90 and 21 cells for rainfall forecasting tasks on Indian weather patterns, providing mean *MSE* losses close to 0.2. Similarly, micro-climate forecasting evaluations in the literature for LSTM models of four layers of stacked LSTMs of 50 cells presented mean *MSE* test results (12, 8, 6, 4, 3, 4, 1, 0.5 hours mean) close to 0.08 (0.06–0.09), while for the BiLSTM model of four layers and 50 cells, this was close to 0.055 (0.02–0.09) [59]. The forecasting results of the stranded LSTM model regarding *MSE* loss are close and less than those provided by four stacked layers of LSTMs. However, the BiLSTM results in this study seem to slightly outperform our *MSE* results. Nevertheless, as the authors of that paper mention, in their validation results, the LSTM model performed 0.04% less well in terms of *RMSE* values than its corresponding BiLSTM model. The above cross-comparison results indicate that the stranded LSTM model outperforms both single-layer LSTM and BiLSTM models with regard to forecasting. Setting BiLSTM models as networks that do not maintain the chronological order of events, the proposed stranded LSTM model can infer better forecasts of less loss, even better than four-layered LSTM models.

The authors conducted their experiments using two distinct cases: One focused on a classifier and the other on a forecaster. Both analyses relied on a previously maintained and experimentally controlled dataset. More specifically, data were collected from an industrial compressor and obtained from the thingsAI platform. These datasets have been previously utilized and tested in the authors' earlier publications. The authors acknowledge the use of pre-existing datasets as a limitation of this study. As future work, the authors propose the exploration of freely available community datasets before undertaking comprehensive statistical analysis, as well as datasets containing attributes from billions of records that the authors have also curated to classify critical apiary events using their stranded LSTM implementation.

The proposed stranded LSTM model is an enhanced version in terms of size, featuring multiple strands and classes and requiring longer training times. This model may offer strategies beyond those discussed in this paper, resulting in more hyperparameters for fine-tuning. However, the weight selection strategies of the stranded LSTM model and the utilization of aggregated inference across a horizontal plane of multiple LSTMs could significantly improve performance in forecasting and classification tasks. This could lead to networks outperforming the stacked LSTM layer models currently in the literature.

## 5. Conclusions

This paper introduces a new model called stranded LSTM. This model employs multiple strands of Long Short-Term Memory (LSTM) cells with varying depths and is designed for classification or prediction tasks on time series sensory data. The strands are categorized into four memory classes corresponding to different time granularities of the data inputs, trying to capture different patterns over time and using different numbers of cells in each strand. The concept of the stranded LSTM model is built upon earlier model proposals. The initial model was the stranded NN, which utilized multiple strands of artificial neural networks (ANNs) composed of fully connected layers of varying depths. Each layer contained fewer neurons than the others, following powers of 2. This model proved effective for classification tasks involving dense sensory measurements, as it identified both short- and long-term patterns, ultimately outperforming conventional LSTM networks and double-layered LSTM networks. Subsequently, an enhanced version of the stranded-NN model, the fuzzy stranded NN, was introduced, which employed an autoencoder based on fuzzy rules for auto-annotating data prior to training. Both of these earlier models focused on classification tasks.

The authors proposed the stranded LSTM model, replacing the neurons from previous stranded models with a series of LSTM cells of arbitrary depth to create a model capable of handling classification and predictive inferences. Each one of the LSTM strands is followed by two neural network layers, with the number of neurons tailored to fit the data output shape—whether for classification or forecasting depth. The strength of the stranded LSTM model lies in its implementation of various LSTM strand selection strategies paired with a class hierarchy of strands, where each class represents different data granularities. These strategies facilitate either weighted inference aggregations or the class-based selection of strands, utilizing training and evaluation parameters such as evaluation loss and training jitter loss.

For inference tasks, the stranded LSTM uses three different selection strategies to generate results: the least loss strategy, the weighted least loss strategy, and the fuzzy least loss strategy. Each strategy incorporates a weight-balancing process to determine the best representative LSTM strand or the best representative strand within each class. When the selection strategy focuses on the best representatives from each class, the final inference result is obtained by calculating a weighted sum of the representatives from each class. The weights for this calculation are derived from the Mean Squared Error (*MSE*) loss for predictors or classification loss for classifiers during the model's evaluation phase. Additionally, the total jitter metric is used during the training validation and evaluation of the model strands.

The authors experimented with their proposed model using various time series datasets for classification and prediction tasks across different scenarios. In their classification experiments, they demonstrated that their model outperformed traditional LSTM models and dual-layered stacked LSTM models ($2 \times 16$ and $2 \times 60$ LSTM models), achieving accuracy improvements from 0.5% to 7%. Additionally, their model could automatically identify the LSTM model with the least loss among a group of models performing the same inference task.

In the forecasting experiments, the authors quantified the significance of their model's aggregated inference policies, showing a reduction in loss from 5% to 7% compared to standard LSTM networks. They also highlighted the importance of composite LSTM models comprising multiple LSTM models and provided an automated selection process for identifying the best candidate model.

The authors highlight the better inference results of their model compared to stacked LSTM models for both forecasting and classification stacks and stacked BiLSTM models

for forecasting tasks. However, their model has limitations, particularly regarding the significantly longer training times associated with the number of LSTM strands it supports. Additionally, they observe that the inference process takes at least four times longer when using the weighted and fuzzy strategies. For future work, the authors plan to experiment further with their proposed models on different datasets and compare them with other models beyond LSTM. They also intend to investigate their model's training time and memory requirements when utilizing cloud services and the inference times when using pre-trained models on edge computing IoT devices.

**Author Contributions:** Conceptualization, S.K.; methodology, C.P.; software, S.K.; validation, S.K. and G.K.; formal analysis, S.K.; investigation, S.K.; resources, C.P.; data curation, G.K.; writing—original draft preparation, S.K.; writing—review and editing, C.P., and G.K.; visualization, S.K.; supervision, C.P.; project administration, C.P. All authors have read and agreed to the published version of the manuscript.

**Funding:** This research received no external funding.

**Data Availability Statement:** No new data were created or analyzed in this study. Data sharing is not applicable to this article.

**Conflicts of Interest:** The authors declare no conflicts of interest.

## Abbreviations

The following abbreviations are used in this manuscript:

| | |
|---|---|
| ANN | Artificial Neural Networks or NNs |
| DL | Deep Learning |
| DTs | Decision Trees |
| EWMA | Exponentially Weighted Moving Average |
| GRU | Gated Recurrent Unit model |
| LSTM | Long Short-Term Memory model |
| *MAE* | Mean Absolute Error |
| *MAPE* | Mean Absolute Percentage Error |
| *MSE* | Mean Squared Error |
| ML | Machine Learning |
| NN | Neural Network model |
| *RMSE* | Root Mean Squared Error |
| RNN | Recurrent Neural Network model |
| SVM | Support Vector Machines |

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
