# Peer review of "Proposed Long Short-Term Memory Model Utilizing Multiple Strands for Enhanced Forecasting and Classification of Sensory Measurements"

_mathematics, doi:10.3390/math13081263_

Round 1
Reviewer 1 Report
Comments and Suggestions for Authors
This paper uses the LSTM neural network for time series prediction. The main contribution of the paper is the use of several LSTM network models, called threads, in an ensemble method to combine different predictions (or classifications) into a single response. The authors claim that the use of different LSTMs with different depths mitigates the problems of overtraining and the vanishing gradient problem that such networks suffer from.
The work seems interesting, and the results improve on those obtained with the use of basic LSTMs, as the authors claim.
In the following, I will make some comments and suggestions about this work.
The method presented is highly dependent on the dataset used. For example, a critical parameter of this type of model is the depth of the LSTM network. The authors propose to use four classes of LSTM threads with increasing depths in powers of 2. But they do not provide information on how to determine (even heuristically) both the number of classes and the number of depths of each class. The solution they propose seems very well adapted to a very specific dataset. I think that the work would be improved by shedding some light on this point.
On lines 135 to 137 we read: "This depth value is a supervised learning transformation of an original time series dataset with vector attributes to an array of continuous input chunks of vector attribute input, annotated with the next vector value as output forecast" what do the authors mean by "supervised learning transformation"?
The authors use metrics based on the so-called MSE jitter for the best model selection part. But the MSE jitter can have a strong dependency on the learning algorithm (a clear example is normal BP vs BP with momentum). Therefore, this affects the thresholds that determine the low, moderate and high jitter values used by the algorithm. Have the authors considered the influence of the learning algorithm used on the obtained MSE jitter values?
The equation on line 300 needs a closing parenthesis.
The acronym EWMA appears on line 240. It should be explained what they mean.
In the comparison of the classification results (section 3.1, table 3 and 4), it is not well explained how the tests are performed. Are the same initial weights used for each network (when possible) or the same seed for the random number generator? Normally for such comparisons, N tests should be performed for each case, so that the results obtained are statistically significant. The authors should clarify this point.
As a conclusion of my review, I would like to point out that the proposed method is a good contribution to ensemble learning, but that there are many parameters that depend on the specific dataset to which it is applied, which increases the number of hyperparameters that must be set to properly apply the technique.
The network models are well explained and the results obtained are better than using other basic models. However, the comparison of the results with other techniques or models can be improved. For example, two very specific datasets belonging to the authors have been used. But the proper way to make a fair comparison is to use standard datasets available to the community so that the comparison can be verified and reproduced. This point is very much the improvement proposed for this work.
Author Response
Reviewer 1 -amendments.
This paper uses the LSTM neural network for time series prediction. The main contribution of the paper is the use of several LSTM network models, called threads, in an ensemble method to combine different predictions (or classifications) into a single response. The authors claim that the use of different LSTMs with different depths mitigates the problems of overtraining and the vanishing gradient problem that such networks suffer from.
The work seems interesting, and the results improve on those obtained with the use of basic LSTMs, as the authors claim.
In the following, I will make some comments and suggestions about this work.
Response: Thank you for your time and effort in reviewing our manuscript. Here, we quote our responses and amendments performed based on your comments. Since we have added more references to our manuscript, and the latexdiff tool cannot handle well the added citations, we have uploaded here a pdf with broken citations, which highlights changes and the new corrected version as the amended manuscript.
Comment 1: The method presented is highly dependent on the dataset used. For example, a critical parameter of this type of model is the depth of the LSTM network. The authors propose to use four classes of LSTM threads with increasing depths in powers of 2. But they do not provide information on how to determine (even heuristically) both the number of classes and the number of depths of each class. The solution they propose seems very well adapted to a very specific dataset. I think that the work would be improved by shedding some light on this point.
Response: Thank you very much for this comment. To address your concerns, an appropriate paragraph has been added, and the paragraphs in lines 141-187 have been rewritten, explaining why the authors used four classes and their corresponding cell depths. Table 1 has also been added to provide insight into the proposed model's automatic selection capabilities. The authors pinpoint the need to provide an automated process of learning, regardless of the data granularity, that will select the appropriate LSTM network or group of LSTM networks via aggregation to perform classification or forecasting on that specific data. On the contrary, the essence of our proposition is to introduce a model that can automatically select depth for forecasting or classification tasks without the need for an expert to give us insights about the nature and characteristics of the dataset..
Comment 2: On lines 135 to 137 we read: "This depth value is a supervised learning transformation of an original time series dataset with vector attributes to an array of continuous input chunks of vector attribute input, annotated with the next vector value as output forecast" what do the authors mean by "supervised learning transformation"?
Response: Paragraph bellow figure 2 has been rewritten for clarity:
All participating strands in the model are trained on data using fixed dataset time depth intervals, which are initially set based on the training scenario. This depth represents a transformation of the original time series dataset, where vector attributes are converted into continuous input chunks. Then each chunk is annotated with the next vector value as the forecast output. This data partitioning serves as a pre-training transformation of the raw dataset. The transformed data are then split into training, validation, and evaluation (test) sets based on the following ratios:
For classification: 70\%-20\%-10\% (training, validation, evaluation).
For forecasting: 70\%-10\%-10\%-10\% (training, validation, evaluation, testing).
During per-strand training and evaluation, the metrics described in Section~\ref{s1:2} are computed for each strand. These values are stored within the model as historical validation and evaluation data. Figure~\ref{fig2} illustrates the layered architecture of each strand, with representative strands from the four model classes.
Comment 3: The authors use metrics based on the so-called MSE jitter for the best model selection part. But the MSE jitter can have a strong dependency on the learning algorithm (a clear example is normal BP vs BP with momentum). Therefore, this affects the thresholds that determine the low, moderate, and high jitter values used by the model. Have the authors considered the influence of the learning algorithm used on the obtained MSE jitter values?
Response: Yes, validation jitter is influenced by poorly selected momentum hyperparameters, especially excessively high ones. The backpropagation (BP) learning rate is also a hyperparameter that can cause overshooting, leading to instability. However, this behavior affects all LSTM strands equally since the learning rate is shared across all strands during BP.
For this reason, the kp parameter in Equation (8) is set to 1 and the expression ln(Kp + ||J_i||) is set down to a maxima of 10 for ||Ji||==20000, to compensate for erroneous jitter values, effectively increasing the importance of evaluation loss over jitter. Nevertheless, if the learning rate spikes due to an adaptive learning rate policy, this will impact validation losses and, consequently, all models. Since all models are trained sequentially on the same dataset, such effects propagate throughout the training process.
Poorly chosen momentum hyperparameter values affect the validation jitter, particularly when set too high. The backpropagation (BP) learning rate is another hyperparameter that can lead to overshooting, which causes instability. However, this issue impacts all LSTM strands equally since the learning rate is shared across all strands during backpropagation.
To address this, the kp parameter in Equation (8) is set to 1, and the expression \( \ln(Kp + ||J_i||) \) has an upper boundary value of 10 for very big jitter values $||J_i||\le20,000$. This adjustment helps mitigate erroneous jitter values by effectively increasing the significance of evaluation loss in comparison to jitter. Nonetheless, if the learning rate experiences spikes due to an adaptive learning rate policy, it will affect validation losses and, consequently, all models. Since all models are trained sequentially on the same dataset, these effects can propagate throughout the training process. An appropriate justification paragraph has been added after Equation 8.
Comment 4: The equation on line 300 needs a closing parenthesis.
The acronym EWMA appears on line 240. It should be explained what they mean.
Response: Parenthesis has been added to the equation on line 300. The authors also made typo syntactical and grammatical corrections (highlighted) to improve their manuscript's readability. The EWMA acronym is explained in its first line of appearance.
Comment 5: In the comparison of the classification results (section 3.1, table 3 and 4), it is not well explained how the tests are performed. Are the same initial weights used for each network (when possible) or the same seed for the random number generator? Normally, for such comparisons, N tests should be performed for each case, so that the results obtained are statistically significant. The authors should clarify this point.
Response: Appropriate amentments have been made in section 3.1, lines 480-505 and an additional paragraph was added explaining in detail how the tests were performed.
Comment 6: As a conclusion of my review, I would like to point out that the proposed method is a good contribution to ensemble learning, but that there are many parameters that depend on the specific dataset to which it is applied, which increases the number of hyperparameters that must be set to properly apply the technique.
Response: Thank you very much for this comment, but we think that it is rather an increase in the model size, including multiple strands and classes. The parameters of different aggregation strategies (weighted, fuzzy), such as Kp and alpha, are certainly an increase but are the least it can be done with respect to the impact of selection strategies that may have to utilize aggregations on a horizontal plane of many LSTMs with respect to providing forecasting and classification tasks. That, in turn, will lead to networks that can perform better than the stacked LSTM layer models that are presented today in the literature. An additional paragraph has been added in the results discussion section (last paragraph), mentioning the trade-off of adding hyperparameters with respect to better classification and forecasting results.
Comment 7: The network models are well explained, and the results obtained are better than using other basic models. However, the comparison of the results with other techniques or models can be improved. For example, two very specific datasets belonging to the authors have been used. But the proper way to make a fair comparison is to use standard datasets available to the community so that the comparison can be verified and reproduced. This point is very much the improvement proposed for this work.
Response: Thank you very much for this comment. However, since our model requires a fair amount of data. There are a few datasets freely available that the authors can with certainty verify that are gathered under controlled conditions with minimum errors and outliers. That is why we have used datasets already controlled and tested by the authors in previous publications. As a generic open-source dataset, we also consider the Mafaulda dataset, and a dataset of bee sounds kept by our laboratory for the classification of queenless, swarming, normal, and weak bees. An appropriate limitations paragraph has been added at the end of section 4.

Reviewer 2 Report
Comments and Suggestions for Authors This is potentially a good solution to time-series analysis, but I have a few concerns with the current version of the manuscript, because of which I cannot recommend acceptance, which is as follows, 1. The experiments focus solely on comparing the proposed model with standard dual-layer LSTM networks. To strengthen the results, the experiments should include comparisons with other RNN models, such as BiLSTM, GRU, BiGRU. 2. The cell counts per class (e.g., 8–32 for Class 1) and the step increments (powers of 2) are empirically motivated but lack theoretical or citation-backed justification. Add references or ablation studies to validate these design choices. 3. The architecture descriptions (e.g., Figure 2 and Figure 3) are dense and mathematically thorough but lack clarity for visual understanding. Consider simplifying the illustrations and adding clear flowcharts to show data input, strand selection, aggregation, and final output. 4. The paper references previous works on stranded-NN and fuzzy stranded-NN models by the same authors. The novelty of using multiple LSTM strands should be clearly distinguished from prior works. What is new about the stranded-LSTM architecture compared to earlier models? 5. There are issues like "Section ?? presents..." in line 99 and two full stops in line 382. The authors should thoroughly revise the manuscript.Author Response
Reviewer 2 - amendments
This is potentially a good solution to time-series analysis, but I have a few concerns with the current version of the manuscript, because of which I cannot recommend acceptance, which is as follows.
Response: Thank you for your time and effort in reviewing our manuscript. Here, we quote our responses and amendments performed based on your comments. Since we have added more references to our manuscript, and the latexdiff tool cannot handle well the added citations, we have uploaded here a pdf with broken citations, which highlights changes and the new corrected version as the amended manuscript.
Comment 1. The experiments focus solely on comparing the proposed model with standard dual-layer LSTM networks. To strengthen the results, the experiments should include comparisons with other RNN models, such as BiLSTM, GRU, BiGRU.
Response: Thank you very much for your comment. An appropriate paragraph has been added in lines 70-76, stating why the authors used LSTM over GRU models in this study. An appropriate paragraph has been added in lines 82-96, indicating the inabilities of BiLSTM and BiGRU models toward forecasting tasks. Furthermore, an additional paragraph has been added in the discussion of the results section, lines 606- 623, with cross-comparison results.
Comment 2. The cell counts per class (e.g., 8–32 for Class 1) and the step increments (powers of 2) are empirically motivated but lack theoretical or citation-backed justification. Add references or ablation studies to validate these design choices.
Response: Thank you very much for this comment. Appropriate justification and citations have been added in lines 141-163, including a new Table 1.
Comment 3. The architecture descriptions (e.g., Figure 2 and Figure 3) are dense and mathematically thorough but lack clarity for visual understanding. Consider simplifying the illustrations and adding clear flowcharts to show data input, strand selection, aggregation, and final output.
Response: Figures 2 and 3 have been simplified both at LSTM strand level and at stranded LSTM model level.
Comment 4. The paper references previous works on stranded-NN and fuzzy stranded-NN models by the same authors. The novelty of using multiple LSTM strands should be clearly distinguished from prior works. What is new about the stranded-LSTM architecture compared to earlier models?
Response: Thank you very much for this comment. The authors propose two models in previous works. The first proposition was the stranded-NN model that used multiple strands of ANNs of fully connected layers of different depths, where each layer included fewer neurons than the others, using powers of 2. This model was successfully used for classification tasks of dense sensory measurements where we were looking for short- and long-term patterns. This model managed to outperform both LSTM networks as well as double-layered LSTM networks. Then, we experimented with another version of the stranded-NN model, the fuzzy-stranded-NN, that managed to auto-annotate data using an autoencoder based on fuzzy rules prior to training. Both previously proposed models have been used for classification tasks. This proposition of the stranded LSTM replaces the neurons of the previous stranded models with LSTM cells of arbitrary cell depth, followed by a set of strand selection strategies. These strategies can either perform weighted inference aggregations or strand selection based on strand training and evaluation parameters of evaluation loss and training jitter loss. The appropriate paragraphs have been added at the beginning of the conclusions section.
Comment 5. There are issues like "Section ?? presents..." in line 99 and two full stops in line 382. The authors should thoroughly revise the manuscript.
Response: The authors fixed the mentioned issues. They have also made typo syntactical and grammatical corrections (highlighted in their manuscript) to improve their manuscript's readability.

Round 2
Reviewer 2 Report
Comments and Suggestions for Authors
The author has addressed most of the previous questions well. However, there is still a minor issue: in Figure 2, the text within the flowchart on the right is sometimes centered and sometimes left-aligned. Please unify it to be centered. A similar situation occurs in Figure 3.
Author Response
Comment: The author has addressed most of the previous questions well. However, there is still a minor issue: in Figure 2, the text within the flowchart on the right is sometimes centered and sometimes left-aligned. Please unify it to be centered. A similar situation occurs in Figure 3.
Response: Thank you again for your time and effort in reviewing our manuscript. We have updated Figures 2 and 3.